# Comorbidity clusters and in-hospital outcomes in patients admitted with acute myocardial infarction in the USA: A national population-based study

Salwa S. Zghebi[1,2]*, Martin K. Rutter[3,4], Louise Y. Sun[5], Waqas Ullah[6], Muhammad Rashid[7,8], Darren M. Ashcroft[9,10], Douglas T. Steinke[9], Stephen Weng[11], Evangelos Kontopantelis[1,12], Mamas A. Mamas[7,8]

1 Centre for Primary Care and Health Services Research, Division of Population Health, Health Services Research and Primary Care, School of Health Sciences, The University of Manchester, Manchester, United Kingdom, 2 Department of Pharmaceutics, Faculty of Pharmacy, University of Tripoli, Tripoli, Libya, 3 Diabetes, Endocrinology & Metabolism Centre, Manchester University NHS Foundation Trust, NIHR Manchester Biomedical Research Centre, Manchester Academic Health Science Centre, Manchester, United Kingdom, 4 Division of Diabetes, Endocrinology & Gastroenterology, School of Medical Sciences, The University of Manchester, Manchester, United Kingdom, 5 Division of Cardiothoracic Anesthesiology, Department of Anesthesiology, Perioperative and Pain Medicine, Stanford University School of Medicine, Palo Alto, California, United States of America, 6 Department of Cardiology, Thomas Jefferson University Hospitals, Philadelphia, Pennsylvania, United States of America, 7 Keele Cardiovascular Research Group, Centre for Prognosis Research, School of Medicine, Keele University, Stoke-on-Trent, United Kingdom, 8 Department of Academic Cardiology, Royal Stoke University Hospital, Stoke-on-Trent, United Kingdom, 9 Centre for Pharmacoepidemiology and Drug Safety, Division of Pharmacy and Optometry, School of Health Sciences, The University of Manchester, Manchester, United Kingdom, 10 NIHR Greater Manchester Patient Safety Research Collaboration (PSRC), The University of Manchester, Manchester, United Kingdom, 11 Development Biostatistics, GSK, Stevenage, United Kingdom, 12 Division of Informatics, Imaging and Data Sciences, School of Health Sciences, The University of Manchester, Manchester, United Kingdom

* salwa.zghebi@manchester.ac.uk

## Abstract

### Background

The prevalence of multimorbidity in patients with acute myocardial infarction (AMI) is increasing. It is unclear whether comorbidities cluster into distinct phenogroups and whether are associated with clinical trajectories.

### Methods

Survey-weighted analysis of the United States Nationwide Inpatient Sample (NIS) for patients admitted with a primary diagnosis of AMI in 2018. In-hospital outcomes included mortality, stroke, bleeding, and coronary revascularisation. Latent class analysis of 21 chronic conditions was used to identify comorbidity classes. Multivariable logistic and linear regressions were fitted for associations between comorbidity classes and outcomes.

**Data Availability Statement:** The third party data underlying the results presented in this study are available upon request from the Healthcare Cost

and Utilization Project (HCUP) - National (Nationwide) Inpatient Sample (NIS) via their website (https://www.hcup-us.ahrq.gov/nisoverview.jsp). All interested researchers can access the data through HCUP directly. The authors of this study are not permitted to share the data or make it publicly available as per the data use agreement with HCUP. The authors did not have any special access privileges to this data.

**Funding:** This study is funded by The University of Manchester as part of the Presidential Fellowship provided to SSZ. The funders had no role in study design, data collection and analysis, decision to publish, or preparation of the manuscript.

**Competing interests:** SSZ, LYS, EK, MKR, DS, DMA MAM, MR declare no competing interests. SW is a currently an employee of GSK. This does not alter our adherence to PLOS ONE policies on sharing data and materials.

## Results

Among 416,655 AMI admissions included in the analysis, mean (±SD) age was 67 (±13) years, 38% were females, and 76% White ethnicity. Overall, hypertension, coronary heart disease (CHD), dyslipidaemia, and diabetes were common comorbidities, but each of the identified five classes (C) included ≥1 predominant comorbidities defining distinct phenogroups: cancer/coagulopathy/liver disease class (C1); least burdened (C2); CHD/dyslipidaemia (largest/referent group, (C3)); pulmonary/valvular/peripheral vascular disease (C4); diabetes/kidney disease/heart failure class (C5). Odds ratio (95% confidence interval [CI]) for mortality ranged between 2.11 (1.89–2.37) in C2 to 5.57 (4.99–6.21) in C1. For major bleeding, OR for C1 was 4.48 (3.78; 5.31); for acute stroke, ORs ranged between 0.75 (0.60; 0.94) in C2 to 2.76 (2.27; 3.35) in C1; for coronary revascularization, ORs ranged between 0.34 (0.32; 0.36) in C1 to 1.41 (1.30; 1.53) in C4.

## Conclusions

We identified distinct comorbidity phenogroups that predicted in-hospital outcomes in patients admitted with AMI. Some conditions overlapped across classes, driven by the high comorbidity burden. Our findings demonstrate the predictive value and potential clinical utility of identifying patients with AMI with specific comorbidity clustering.

## Introduction

Coronary heart disease (CHD) is the leading cause of mortality worldwide [1, 2], with over 800,000 patients sustaining an acute myocardial infarction (AMI) each year in the US [1]. Multimorbidity (or comorbidity) is defined as the co-existence of two or more comorbidities in the same individual [3]. The number of people living with multiple long-term conditions has increased in recent years driven by increased life expectancy and improved healthcare [3]. In the US, the prevalence of multimorbidity and affects more than one quarter of adults [4, 5]. Among patients with incident cardiovascular disease (CVD), 81.1% have at least one comorbidity and the proportion of patients who develop multiple comorbidities is increasing over time [6]. Studying multimorbidity patterns informs clinical guidelines, healthcare policy developers and healthcare professionals to better understand the holistic care needs of patients. Multimorbidity is multifaceted with links to polypharmacy, low medication adherence, adverse clinical outcomes including readmissions, poor quality of life and life satisfaction [7, 8].

Despite the current increasing prevalence of multimorbidity in people with CVD and the potential impact of associated outcomes, identifying multimorbidity patterns/clustering is limited. Latent class analysis (LCA) can be used to identify such clusters, identifying classes of 'similar' individuals grouped based on a set of observed variables, such as comorbidities [9–11], and it performs better than other clustering methods [12]. Some past studies on US cardiovascular admissions examined comorbidities but not their detailed patterns or in terms of multimorbidity burden or looked at latent subgroups in the prediction adverse outcomes in people admitted with AMI [13–16].

In this study, we used US nationwide data to identify patients with AMI as the primary cause of hospitalization. We aimed to 1) examine comorbidity burden and how it may cluster in different groups using data on 21 chronic conditions; 2) describe the socio-demographic

characteristics of the identified latent comorbidity groups; and 3) examine associations between the comorbidity classes and in-hospital outcomes.

## Methods

### Data source

Data were derived from the US Nationwide Inpatient Sample (NIS) which is the largest all-payer data on inpatient stays from all US hospitals participating in the Healthcare Cost and Utilization Project (HCUP). Sponsored by the US Agency for Healthcare Research and Quality (AHRQ), the NIS data is the largest available all-payer data on inpatient stays from US states participating in the Healthcare Cost and Utilization Project (HCUP), covering over 97% of the US population [17]. The design of NIS approximates a 20% stratified sample of all admissions from long-term acute care hospitals and community hospitals, except rehabilitation [17]. The NIS provides anonymized data on primary and secondary hospitalization diagnoses from over 7 million annual inpatient stays between 2004 and 2018, recorded in discharge abstracts [17, 18]. AHRQ sampling weights were applied for each admission for national level estimates. We used modified weights in all analyses to take the NIS sampling design change in 2012 into account. Further database documentation is available online [19].

### Study design and study population

This was a retrospective cohort study including data for all individuals aged ≥35 years with acute myocardial infarction (AMI) (International Classification of Diseases (ICD)-Tenth Revision (ICD-10) codes I21*, I22*) as the primary discharge diagnosis between January and December 2018.

### Data variables and outcomes

Variables included were age, sex, race (White, Black, Hispanic, Asian or Pacific Islander, Native American, Other, unknown), median household income quartiles for patient's ZIP Code, weekend admission, total hospital charges, length of hospital stay (LOS), and hospital region. Admission records with missing age, sex, LOS, total costs, elective, weekend admissions, or death status were excluded (N = 685, 0.8%). Other missing variables, such as race were assigned to a separate 'unknown' category.

We extracted data on 21 comorbidities recorded in the admission records using the HCUP Elixhauser comorbidity software ICD-10 codes [20]; diabetes (with or without complications) (DM), peripheral vascular disease (PVD), heart failure (HF), valvular disease (VD) (including stenosis, insufficiency, congenital, prosthetic, or rheumatic disorders of heart and pulmonary valves), hypertension (HTN) (including gestational HTN), dyslipidaemia, coronary heart disease (CHD), anaemias, atrial fibrillation or flutter (AF), coagulopathy, depression, obesity, chronic kidney disease (CKD), hypothyroidism, transient ischemic attack (TIA)/ stroke, rheumatoid arthritis (RA)/collagen vascular diseases, liver disease, cancer, weight loss, psychoses, and chronic obstructive pulmonary disease (COPD).

In-hospital outcomes of interest were all-cause mortality, acute ischemic stroke, major bleeding (including haemorrhagic stroke), procedure-related bleeding, cardiac tamponade, use of assist device or intra-aortic balloon pump (IABP), coronary artery bypass graft (CABG), and percutaneous coronary intervention (PCI), LOS (days), and total hospital costs (S1 Table).

## Statistical analysis

Categorical variables, described as frequencies with percentages, were compared using Pearson's Chi-squared test. Continuous variables, described as mean and standard deviation (SD) or median and interquartile range (IQR) (as appropriate as per the normal distribution of the variable), were compared using the Mann-Whitney U test for differences across 2 groups (such as gender), or using the Kruskal-Wallis test (non-normally distributed variables) or one-way ANOVA (normally distributed variables) for differences across >2 groups (such as latent classes). Nominal (exploratory) P-values were estimated and reported.

Generalized structural equation modelling (gsem) was used to identify categorical unobserved (latent) classes [21]. Latent class analysis relates a set of observed categorical variables (in this case the 21 comorbidities) to a set of latent variables, thus assigning the population to subgroups of 'similar' individuals referred to as 'latent classes' [9–11]. LCA is a common approach to identify comorbidity clusters and is reportedly superior to existing methods [12]. As suggested previously, the number of latent classes can be decided by comparing the values of Akaike's information criterion (AIC) and Schwarz's Bayesian information criterion (BIC) of models based on different numbers of classes, where smaller AIC and BIC values are better and indicate the optimal number of classes [9, 22, 23]. Following assessment of models with 2–8 classes, AIC and BIC values decreased marginally with increasing number of classes from two to six (S2 Table). Given the trivial differences of AIC/BIC values (0.11%/0.10%) between the 5-class/6-class models, the 5-class model was chosen as a reasonable compromise, showing informative separation of the examined 21 comorbidities with a smaller number of classes. Latent class marginal probabilities (95% CI) were estimated.

Multivariable logistic regression models estimated odds ratios (OR) and 95% confidence intervals (CI) for likelihood of all binary outcomes, such as mortality and PCI. Negative binomial regression (selected over Poisson regression given the dispersion of LOS data) estimated incidence rate ratios (IRR) and 95%CIs for predictors of count outcome (LOS). Multivariable linear regression estimated regression coefficients and 95%CIs for predictors of total hospital charges. The largest (most common) class was used as the reference group in all regression models to assess the risk of outcomes in comparison to AMI patients with the most common comorbidity burden profile. Analyses were survey-weighted using the provided sampling weights to produce a nationally representative estimate of the entire US population of hospitalized patients.

## Results

### Comorbidity latent classes

Overall, mean (±SD) age was 67 (±13) years, 38% were females, and 76% were White and 12% were Black (Table 1). Patients in Class 2 were the youngest (62 ± 13 years) and had the highest male proportion (66%, 66–67%) and the highest proportion on Medicaid as the primary expected payer (10%, 10–11%), and the lowest hospital costs ($59,855 (IQR 57,402) compared to patients in other classes (Table 1 and S1 Fig). On the other hand, the COPD/VD/PVD group (Class 4) included the oldest patients (75 ± 11 years) and fewest Hispanic individuals among all patients hospitalized with AMI, while the diabetes/CKD/HF group (Class 5) had the lowest proportion of White patients (70%) and highest of Black patients (17%). By both age and sex, 64% of females vs. 51% of males in Class 4 were aged 75+years (S1 Fig). Results of significance testing for differences across the latent classes are summarized in S3 Table.

**Table 1. Baseline characteristics of the emerged comorbidity latent classes.**

| | Overall | Class 1 (Cancer/ coagulopathy/ liver) | Class 2 (Least burdened) | Class 3 (CHD/ dyslipidaemia) | Class 4 (COPD/ VD/PVD) | Class 5 (DM/ CKD/HF) |
|---|---|---|---|---|---|---|
| Weighted admissions, N (%) | 416,655 | **39,640 (9.5%)** | 85,455 (20.5%) | **173,825 (42%)** | 47,525 (11%) | 70,210 (17%) |
| Admissions per 100,000 population | 127.4 | 12.1 | 26.1 | 53.1 | 14.5 | 21.5 |
| Age (years), mean ±SD | 67±13 | 71±14 | **62±13** | 65±12 | **75±11** | 72±11 |
| Age categories, % (95% CI) | | | | | | |
| 35-44y | 4.3 (4.11; 4.4) | 3.4 (3.0; 3.8) | 8.3 (7.9; 8.8) | 4.5 (4.3; 4.8) | 0.9 (0.7; 1.1) | 1.3 (1.1; 1.5) |
| 45-54y | 13.9 (13.6; 14.1) | 10.1 (9.5; 10.8) | 22.3 (21.7; 23.0) | 16.1 (15.7; 16.5) | 4.0 (3.7; 4.4) | 6.8 (6.4; 7.2) |
| 55-64y | 24.8 (24.5; 25.1) | 20.2 (19.3; 21.1) | 30.1 (29.5; 30.8) | 28.9 (28.4; 29.4) | 13.4 (12.8; 14.1) | 18.3 (17.7; 19.0) |
| 65-74y | 26.5 (26.2; 26.8) | 23.9 (23.0; 24.8) | 21.6 (22.0; 22.2) | 28.0 (27.5; 28.4) | 25.3 (24.4; 26.2) | 31.4 (30.6; 32.2) |
| 75+ y | 30.6 (30.3; 30.9) | 42.5 (41.4; 43.6) | 17.6 (17.1; 18.2) | 22.5 (22.0; 22.9) | 56.3 (55.3; 57.3) | 42.2 (41.4; 43.0) |
| Sex, % (95% CI) | | | | | | |
| Male | 62.4 (62.0; 62.7) | 55 (54.3; 56.5) | **66 (65.5; 66.9)** | 65 (64.9; 65.9) | 57 (55.6; 57.6) | 58 (57.3; 59.0) |
| Female | 37.6 (37.3; 38.0) | **45 (43.5; 45.7)** | 34 (33.1; 34.5) | 35 (34.2; 35.2) | 43 (42.4; 44.4) | 42 (41.0; 42.7) |
| Race, % (95% CI) | | | | | | |
| White | 75.6 (75.3; 75.9) | 72.9 (71.9; 73.9) | 77.9 (77.3; 78.5) | 75.8 (75.3; 76.2) | 82.0 (81.2; 82.8) | 69.6 (68.8; 70.4) |
| Black | 12.1 (11.9; 12.4) | 15.2 (14.4; 16.0) | 9.6 (9.2; 10.0) | 11.4 (11.0; 11.7) | 9.4 (8.8; 10.0) | **17.2 (16.6; 17.9)** |
| Hispanic | 4.9 (4.7; 5.0) | 4.7 (4.27; 5.21) | 4.7 (4.4; 5.0) | 5.2 (5.0; 5.4) | **2.9 (2.6; 3.2)** | 5.8 (5.4; 6.2) |
| Asian/Pacific Islander | 1.4 (1.3; 1.5) | 1.2 (0.96; 1.4) | 1.5 (1.3; 1.7) | 1.4 (1.3; 1.5) | 1.1 (0.9; 1.3) | 1.7 (1.5; 2.0) |
| Native American | 0.4 (0.3; 0.4) | 0.3 (0.2; 0.5) | 0.3 (0.3; 0.4) | 0.4 (0.3; 0.4) | 0.3 (0.2; 0.4) | 0.5 (0.4; 0.6) |
| Other | 2.5 (2.4; 2.7) | 2.7 (2.3; 3.0) | 2.5 (2.3; 2.8) | 2.8 (2.6; 2.9) | 1.74 (1.5; 2.0) | 2.5 (2.2; 2.8) |
| Unknown | 3.1 (3.0; 3.2) | 3.0 (2.7; 3.4) | 3.5 (3.3; 3.8) | 3.2 (3.0; 3.3) | 2.7 (2.4; 3.0) | 2.7 (2.4; 2.9) |
| STEMI, % (95% CI) | 27.0 (26.7; 27.3) | 24.36 (23.4; 25.3) | **41.19 (40.5; 41.9)** | 28.93 (28.5; 29.4) | 18.4 (17.7; 19.2) | **12.43 (11.9; 13.0)** |
| Median household income quartiles, % (95% CI) | | | | | | |
| $1-$43,999 | 30.3 (30.0; 30.6) | 32.9 (31.9; 34.0) | **27.5 (26.8; 28.1)** | 29.8 (29.3; 30.3) | 29.9 (29.0; 30.9) | **34.0 (33.2; 34.8)** |
| $44,000-$55,999 | 28.9 (28.6; 29.2) | **29.3 (28.4; 30.4)** | **28.4 (27.7; 29.0)** | 29.0 (28.5; 29.4) | **29.3 (28.4; 30.3)** | 29.0 (28.1; 29.6) |
| $56,000-$73,999 | 22.9 (22.6; 23.2) | **20.6 (19.7; 21.5)** | **24.1 (23.5; 24.8)** | 23.4 (22.9; 23.8) | 22.5 (28.4; 30.3) | 21.6 (21.0; 22.3) |
| ≥ $74,000 | 16.5 (16.2; 16.8) | 15.7 (14.9; 16.5) | **18.5 (17.9; 19.1)** | 16.5 (16.1; 16.9) | 16.9 (16.2; 17.7) | **14.3 (13.8; 14.9)** |
| Unknown | 1.4 (1.4; 1.5) | 1.5 (1.2; 1.8) | **1.6 (1.4; 1.8)** | 1.4 (1.3; 1.6) | 1.4 (1.2; 1.6) | **1.2 (1.0; 1.4)** |
| Primary Payer, % (95% CI) | | | | | | |
| Medicare | 57.9 (57.6; 58.2) | 67.0 (66.0; 68.1) | **38.1 (37.4; 38.9)** | 51.4 (50.9; 51.9) | **79.8 (79.0; 80.6)** | 78.0 (77.3; 78.7) |
| Medicaid | 8.6 (8.4; 8.7) | 9.2 (8.6; 9.8) | **10.3 (9.8; 10.7)** | 9.2 (8.9; 9.5) | **5.1 (4.7; 5.6)** | 6.9 (6.5; 7.3) |
| Private | 25.6 (25.3; 25.9) | 16.5 (15.7; 17.4) | **39.1 (38.4; 39.8)** | 30.7 (30.3; 31.2) | **11.3 (10.7; 11.9)** | 11.4 (10.9; 12.0) |
| Weekend admission, % | 26.3 (26.0; 26.6) | 26.9 (25.9; 27.9) | **27.6 (27.0; 28.3)** | 26.2 (25.8; 26.7) | 25.3 (24.5; 26.2) | **25.2 (24.5; 25.9)** |

*(Continued)*

**Table 1.** (Continued)

| | Overall | Class 1 (Cancer/ coagulopathy/ liver) | Class 2 (Least burdened) | Class 3 (CHD/ dyslipidaemia) | Class 4 (COPD/ VD/PVD) | Class 5 (DM/ CKD/HF) |
|---|---|---|---|---|---|---|
| Length of stay LOS (days), median (IQR) | 3 (3) | 4 (6) | **2 (2)** | **2 (2)** | 4 (5) | **5 (5)** |
| Total hospital charges ($), median (IQR) | 64,758 (74,632) | 60,427 (102,851) | **59,855 (57,402)** | 66,270 (66,567) | 65,168 (94,639) | **69,939 (100,440)** |
| Region of hospital, % (95% CI) | | | | | | |
| Northeast | 25.2 (25.0; 25.5) | **28.9 (27.9; 29.9)** | 27.0 (26.4; 27.7) | 24.2 (23.7; 24.6) | 25.3 (24.5; 26.2) | **23.5 (22.8; 24.2)** |
| Midwest | 32.1 (31.8; 32.4) | **28.2 (27.2; 29.2)** | 30.8 (30.1; 31.5) | 32.8 (32.3; 33.3) | 33.2 (32.2; 34.1) | **33.3 (32.5; 34.1)** |
| South | 42.7 (42.4; 43.0) | 42.9 (41.8; 44.0) | 42.2 (41.5; 42.9) | 43.0 (42.5; 43.5) | **41.5 (40.5; 42.5)** | 43.2 (42.4; 44.0) |

Significance testing: Nominal (exploratory) P-values across the five latent classes were all significant (p<0.001).

CHD: coronary heart disease; CI: confidence interval; CKD: chronic kidney disease; COPD: chronic obstructive pulmonary disease; DM: diabetes; HF: heart failure; IQR: interquartile range; PVD: peripheral vascular disease; SD: standard deviation; STEMI: ST segment elevation myocardial infarction; VD: valvular disease.

A total of 416,655 weighted AMI admissions (N = 83,331 unweighted), corresponding to 127.4 admissions per 100,000 population, were included. More than 99% of patients had ≥1 comorbidity; mean (±SD) of 4.7 (±2.1) comorbidities. Out of the five latent classes that emerged, latent Class 3 was the largest (42%) with highest probability of patient membership (36%), while Class 1 was the smallest (9.5%) with lowest probability of membership (10.8%). The comorbidity profile for the full cohort, across the five latent classes and by sex is described in Table 2, Fig 1 and S2 Fig. The most prevalent comorbidities across all classes were HTN, CHD, dyslipidaemia, and diabetes. However, each class included the highest prevalence of ≥1 unique comorbidity defining five distinct phenogroups: patients in Class 1 had the highest prevalence of cancer, coagulopathy, and liver disease; Class 2 included the youngest and least burdened patients but with highest proportion of ST segment elevation myocardial infarction (STEMI) (41%); CHD and dyslipidaemia in Class 3; COPD, VD, PVD in Class 4; and diabetes, CKD, HF in Class 5. Patients in Class 5 had overall the highest comorbidity burden compared to patients in other classes. Heat maps for comorbidity profile per class are presented in Fig 2.

## Associations between latent classes and in-hospital outcomes

**In-hospital mortality.** Overall, 4.4% of the cohort died in hospital (S4 Table). There was large variation in mortality rate; from 11% of patients in Class 1 to 1.8% in Class 3 (S3 Fig). Patients from all classes, including the youngest/least burdened phenogroup (Class 2), were between 2- and 5-fold higher odds of in-hospital mortality than patients in Class 3 (Table 3 and S5 Table, Fig 3).

**Major bleeding.** Overall, 1.7% had major bleeding with rates ranging between 0.8% in Classes 2 and 3 to 4% in Class 1 (S4 Table, S3 Fig). The ORs (95% CI) for major bleeding were 3.14 (95%CI: 2.63–3.74) in the COPD/PVD/VD (Class 4), and 4.48 (3.78–5.31) in the cancer/ coagulopathy/liver group (Class 1), when compared to Class 3.

**Acute ischemic stroke.** Overall, 1.1% had stroke with rates ranging between 0.6% in Class 2 and 2.4% in Class 1 (S4 Table). Patients in classes 1, 4 and 5 were at higher risks (ORs ranging between 1.71 (1.42; 2.06) to 2.76 (2.27; 3.35)), whereas patients in Class 2 were less likely to experience stroke (OR = 0.75, 0.60–0.94), than Class 3.

**Table 2. Comorbidity classes emerging from latent class analysis of comorbidity profile of patients admitted with AMI in 2018.**

| | Overall | Class 1 (Cancer/ coagulopathy/ liver) | Class 2 (Least burdened) | Class 3 (CHD/ dyslipidaemia) | Class 4 (COPD/ VD/PVD) | Class 5 (DM/ CKD/HF) |
|---|---|---|---|---|---|---|
| Weighted No. of admissions, N (%) | 416,655 | **39,640 (9.5%)** | 85,455 (20.5%) | **173,825 (42%)** | 47,525 (11%) | 70,210 (17%) |
| Patient membership, % probability (95% CI) | NA | 10.8 (10.0–11.6) | 22.7 (20.9–24.6) | **35.6 (33.8–37.5)** | 13.4 (12.4–14.4) | 17.5 (16.4–18.7) |
| Comorbidities, % proportion (95% CI) | | | | | | |
| HTN (incl. gestational HTN) | 81.8 (81.5; 82.1) | 66.5 (65.5; 67.5) | **40.4 (39.7; 41.2)** | 95.0 (94.8; 95.2) | 94.4 (93.9; 94.8) | **99.6 (99.5; 99.7)** |
| Coronary heart disease (CHD) | 80.8 (80.5; 81.1) | **46.6 (45.5; 47.7)** | 67.2 (66.5; 67.9) | **90.6 (90.3; 90.9)** | 88.5 (87.8; 89.1) | 87.3 (86.8; 87.9) |
| Dyslipidaemia | 68.5 (68.1; 68.8) | **5.6 (5.1; 6.1)** | 35.2 (34.5; 35.9) | **92.6 (92.3; 92.9)** | 83.1 (82.4; 83.9) | 74.7 (74.0; 75.4) |
| Diabetes (DM) | 40.9 (40.5; 41.2) | 23.8 (22.8; 24.7) | **5.8 (5.5; 6.2)** | 51.2 (50.6; 51.7) | 7.1 (6.6; 7.7) | **90.7 (90.2; 91.1)** |
| Heart failure (HF) | 34.2 (33.9; 34.5) | 61.7 (60.6; 62.7) | **5.5 (5.1; 5.8)** | 18.1 (17.7; 18.5) | 63.4 (62.5; 64.4) | **73.7 (72.9; 74.4)** |
| Chronic obstructive pulmonary disease (COPD) | 22.5 (22.2; 22.8) | 32.7 (31.7; 33.8) | **11.3 (10.8; 11.8)** | 16.9 (16.5; 17.3) | **40.7 (39.7; 41.7)** | 32.0 (31.2; 32.8) |
| Chronic kidney disease (CKD) | 22.1 (21.8; 22.4) | 25.0 (24.0; 25.9) | **0** | 5.9 (5.7; 6.2) | 35.1 (34.2; 36.1) | **78.6 (77.9; 79.3)** |
| Obesity | 20.9 (20.7; 21.2) | 10.2 (9.5; 10.9) | 10.2 (9.7; 10.7) | 29.2 (28.7; 29.6) | **3.4 (3.1; 3.8)** | **31.5 (30.7; 32.3)** |
| Anaemias | 16.0 (15.7; 16.2) | 24.9 (24.0; 25.9) | **2.7 (2.5; 3.0)** | 3.6 (3.4; 3.8) | 26.6 (25.7; 27.5) | **50.6 (49.8; 51.5)** |
| Valvular disease (VD) | 15.3 (15.0; 15.5) | 16.8 (16.0; 17.7) | **5.5 (5.2; 5.8)** | 7.0 (6.7; 7.2) | **45.7 (44.7; 46.7)** | 26.4 (25.7; 27.1) |
| Hypothyroidism | 12.4 (12.1; 12.6) | 11.1 (10.4; 11.8) | **7.2 (6.8; 7.6)** | 11.0 (10.6; 11.3) | **18.9 (18.1; 19.7)** | 18.5 (17.8; 19.1) |
| Atrial fibrillation (AF)/flutter | 11.2 (11.0; 11.4) | 22.6 (21.7; 23.5) | **3.3 (3.0; 3.5)** | 5.1 (4.9; 5.3) | **27.8 (26.9; 28.7)** | 18.3 (17.7; 19.0) |
| Peripheral vascular disease (PVD) | 9.5 (9.3; 9.74) | 6.2 (5.7; 6.8) | **3.2 (2.9; 3.4)** | 4.5 (4.3; 4.7) | **40.4 (39.5; 41.4)** | 10.8 (10.3; 11.4) |
| Depression | 9.4 (9.2; 9.6) | 8.0 (7.4; 8.6) | **5.0 (4.7; 5.4)** | 9.9 (9.6; 10.2) | 12.2 (11.5; 12.9) | **12.4 (11.9; 13.0)** |
| TIA/Stroke | 7.9 (7.8; 8.1) | 4.1 (3.67; 4.55) | **1.7 (1.5; 1.9)** | 6.6 (6.31; 6.8) | **18.4 (17.7; 19.2)** | 14.0 (13.5; 14.6) |
| Coagulopathy | 5.9 (5.8; 6.1) | **15.0 (14.3; 15.8)** | **1.1 (1.0; 1.3)** | 3.1 (2.90; 3.3) | 11.1 (10.5; 11.7) | 10.3 (9.8; 10.8) |
| Weight loss | 3.0 (2.9; 3.1) | **14.2 (13.4; 15.0)** | 0.4 (0.4; 0.6) | **0.2 (0.2; 0.23)** | 7.4 (6.9; 7.9) | 3.8 (3.5; 4.1) |
| Cancer | 2.8 (2.7; 3.0) | **9.7 (9.1; 10.4)** | **1.0 (0.8; 1.1)** | 1.2 (1.1; 1.3) | 5.8 (5.4; 6.3) | 3.4 (3.1; 3.7) |
| RA & collagen vascular diseases | 2.6 (2.5; 2.7) | 3.7 (3.3; 4.1) | **1.9 (1.7; 2.1)** | **1.9 (1.7; 2.0)** | **5.6 (5.1; 6.1)** | 2.9 (2.6; 3.2) |
| Liver disease | 2.2 (2.1; 2.3) | **6.5 (6.0; 7.1)** | **0.5 (0.4; 0.6)** | 1.8 (1.7; 2.0) | 1.7 (1.4; 1.9) | 3.5 (3.2; 3.8) |
| Psychoses | 2.1 (2.0; 2.2) | **4.5 (4.0; 4.9)** | 1.3 (1.2; 1.5) | 2.2 (2.0; 2.3) | **0.8 (0.6; 1.0)** | 2.2 (2.0; 2.4) |

Significance testing: Nominal (exploratory) P-values across the five latent classes were all significant (p<0.001).

Top and bottom prevalence per comorbidity presented in bold.

HTN: hypertension; RA: rheumatoid arthritis; TIA: transient ischemic attack.

**Other outcomes.** Overall, 51% and 8.1% received PCI and CABG, respectively (S4 Table). Approximately, 30% of patients in the Class 1 vs. 63% of Class 2 underwent PCI (S3 Fig). All patients, except Class 2, had lower odds for receipt of PCI and 1-day longer stays, in comparison to Class 3. Estimates for the other outcomes are described in Table 3, S5–S7 Tables. S4 Fig summarizes the predictive margins of probabilities of binary outcomes.

## Discussion

### Main findings

Using a representative cohort of 416,655 AMI admissions from a national US inpatient database, we identified distinct phenogroups defined by varying comorbidity profiles via latent classes. While a few comorbidities were dominant across all classes, such as HTN and CHD, each class had the highest prevalence of ≥1 specific comorbidity, resulting in five unique

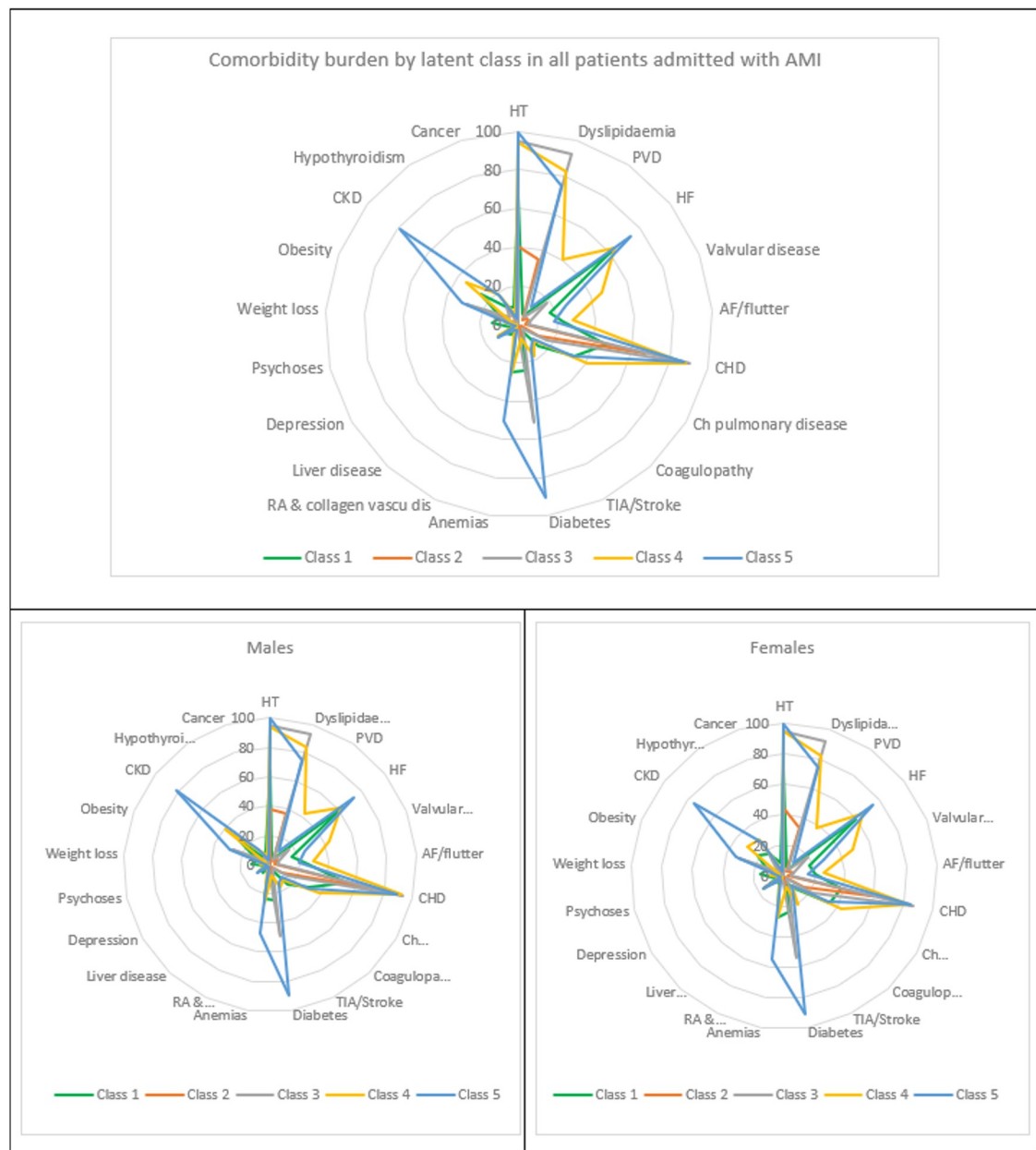

**Fig 1. Radar charts for percentage prevalence of comorbidities per latent class, overall and by sex.** Radar charts present the percentage prevalence of comorbidities per latent class ranging from 0% (chart centre) to 100%.

phenogroups. The youngest/least burdened phenogroup (Class 2) comprised primarily males and had the highest proportion of STEMI admissions. By contrast, the oldest class (Class 4) had the lowest proportion of Hispanic patients and lowest prevalence of obesity. The phenogroups identified have very different clinical outcomes, with those in the cancer/coagulopathy/liver disease class (Class 1) having the greatest risk of in-hospital mortality with 5-fold higher odds compared to patients in Class 3 characterised by CHD/dyslipidaemia. We also report differences in other relevant outcomes including major bleeding (between 0.8%-4%) and in-hospital stroke (0.6%-2%). Finally, we observe that there is heterogeneity in revascularisation, with PCI rates varying from 30% to 63%.

| | Overall | | | | | Males | | | | | Females | | | | |
|---|---|---|---|---|---|---|---|---|---|---|---|---|---|---|---|
| | Class 1 | Class 2 | Class 3 | Class 4 | Class 5 | Class 1 | Class 2 | Class 3 | Class 4 | Class 5 | Class 1 | Class 2 | Class 3 | Class 4 | Class 5 |
| HTN | 66.5 | 40.4 | 95.0 | 94.4 | 99.6 | 64.5 | 38.5 | 95.1 | 94.1 | 99.7 | 69.0 | 44.3 | 94.9 | 94.7 | 99.5 |
| Dyslipidaemia | 5.6 | 35.2 | 92.6 | 83.1 | 74.7 | 5.5 | 36.3 | 93.2 | 84.2 | 75.2 | 5.7 | 33.2 | 91.4 | 81.8 | 74.1 |
| PVD | 6.2 | 3.2 | 4.5 | 40.4 | 10.8 | 6.8 | 3.1 | 4.3 | 42.8 | 11.2 | 5.5 | 3.3 | 4.8 | 37.4 | 10.4 |
| HF | 61.7 | 5.5 | 18.1 | 63.4 | 73.7 | 63.3 | 5.9 | 17.2 | 62.1 | 73.1 | 59.7 | 4.5 | 19.8 | 65.2 | 74.5 |
| Valvular disease | 16.8 | 5.5 | 7.0 | 45.7 | 26.4 | 16.3 | 4.9 | 6.3 | 43.7 | 25.2 | 17.5 | 6.7 | 8.1 | 48.3 | 28.0 |
| AF/flutter | 22.6 | 3.3 | 5.1 | 27.8 | 18.3 | 24.1 | 3.5 | 5.5 | 29.3 | 20.1 | 20.7 | 2.7 | 4.3 | 25.8 | 15.9 |
| CHD | 46.6 | 67.2 | 90.6 | 88.5 | 87.3 | 53.4 | 71.6 | 92.8 | 91.8 | 89.4 | 38.1 | 58.6 | 86.3 | 84.1 | 84.5 |
| COPD | 32.7 | 11.3 | 16.9 | 40.7 | 32.0 | 31.1 | 9.7 | 14.4 | 39.0 | 31.4 | 34.8 | 14.4 | 21.6 | 42.9 | 32.9 |
| Coagulopathy | 15.0 | 1.1 | 3.1 | 11.1 | 10.3 | 18.2 | 1.3 | 3.5 | 13.3 | 11.9 | 11.1 | 0.7 | 2.3 | 8.2 | 8.2 |
| TIA/Stroke | 4.1 | 1.7 | 6.6 | 18.4 | 14.0 | 3.8 | 1.5 | 5.6 | 17.1 | 13.3 | 4.5 | 2.1 | 8.3 | 20.2 | 15.1 |
| Diabetes | 23.8 | 5.8 | 51.2 | 7.1 | 90.7 | 24.2 | 5.5 | 49.6 | 7.3 | 90.3 | 23.2 | 6.3 | 54.1 | 7.0 | 91.2 |
| Anemias | 24.9 | 2.7 | 3.6 | 26.6 | 50.6 | 22.8 | 2.2 | 2.8 | 25.9 | 47.6 | 27.5 | 3.7 | 4.9 | 27.4 | 54.8 |
| RA/collag vas dis | 3.7 | 1.9 | 1.9 | 5.6 | 2.9 | 1.6 | 1.0 | 1.2 | 3.5 | 1.8 | 6.3 | 3.6 | 3.1 | 8.3 | 4.3 |
| Liver disease | 6.5 | 0.5 | 1.8 | 1.7 | 3.5 | 7.7 | 0.5 | 1.9 | 1.9 | 3.9 | 5.0 | 0.4 | 1.7 | 1.3 | 2.9 |
| Depression | 8.0 | 5.0 | 9.9 | 12.2 | 12.4 | 5.7 | 3.5 | 7.5 | 9.1 | 10.1 | 10.8 | 8.1 | 14.5 | 16.2 | 15.6 |
| Psychoses | 4.5 | 1.3 | 2.2 | 0.8 | 2.2 | 3.8 | 1.2 | 1.7 | 0.6 | 1.9 | 5.3 | 1.6 | 3.0 | 1.0 | 2.6 |
| Weight loss | 14.2 | 0.4 | 0.2 | 7.4 | 3.8 | 13.0 | 0.3 | 0.2 | 6.4 | 3.7 | 15.7 | 0.6 | 0.3 | 8.7 | 3.9 |
| Obesity | 10.2 | 10.2 | 29.2 | 3.4 | 31.5 | 9.4 | 9.8 | 27.8 | 3.4 | 29.5 | 11.2 | 11.0 | 31.8 | 3.5 | 34.3 |
| CKD | 25.0 | 0 | 5.9 | 35.1 | 78.6 | 27.1 | 0 | 6.4 | 38.6 | 81.0 | 22.3 | 0 | 5.0 | 30.6 | 75.2 |
| Hypothyroidism | 11.1 | 7.2 | 11.0 | 18.9 | 18.5 | 5.8 | 3.7 | 6.5 | 11.6 | 12.8 | 17.7 | 14.0 | 19.3 | 28.3 | 26.4 |
| Cancer | 9.7 | 1.0 | 1.2 | 5.8 | 3.4 | 10.5 | 1.0 | 1.2 | 6.8 | 3.7 | 8.7 | 0.9 | 1.1 | 4.6 | 2.9 |

Lowest — Mid 25% — Highest

**Fig 2. Heat maps for percentage prevalence of comorbidities per latent class.** Based on cut-off value of 25% to resemble the reported prevalence of multimorbidity in US adults. HTN: hypertension; PVD: peripheral vascular disease; HF: heart failure; AF: atrial fibrillation; CHD: coronary heart disease; COPD: chronic obstructive pulmonary disease; TIA: transient ischemic attack; RA/collag vas dis: rheumatoid arthritis/collagen vascular disease; CKD: chronic kidney disease.

## Findings in relation to literature

**Comorbidity burden in patients with AMI.** The number of people living with comorbidities is increasing worldwide. In the US, the prevalence of multimorbidity is growing and affects >25% of adults [4, 5], and it increases with age among patients with CVD [24]. We

**Table 3. Odds ratios (95% CI) for risks of in-hospital outcomes in patients admitted with AMI in 2018.**

| | Class 1 (Cancer/ coagulopathy/ liver) | Class 2 (Least burdened) | Class 3^ (CHD/ dyslipidaemia) | Class 4 (COPD/VD/ PVD) | Class 5 (DM/CKD/ HF) |
|---|---|---|---|---|---|
| In-hospital death | 5.57 (4.99; 6.21) | 2.11 (1.89; 2.37) | Ref | 2.85 (2.54; 3.21) | 2.89 (2.60; 3.22) |
| Major bleeding* | 4.48 (3.78; 5.31) | 1.04 (0.85; 1.27) | Ref | 3.14 (2.63; 3.74) | 3.20 (2.73; 3.75) |
| Acute ischemic stroke | 2.76 (2.27; 3.35) | 0.75 (0.60; 0.94) | Ref | 2.00 (1.63; 2.46) | 1.71 (1.42; 2.06) |
| Procedure-related bleeding | 2.09 (1.33; 3.28) | 1.09 (0.71; 1.65) | Ref | 2.25 (1.48; 3.44) | 0.94 (0.58; 1.53) |
| Cardiac tamponade | 4.28 (2.64; 6.94) | 1.18 (0.70; 2.01) | Ref | 3.70 (2.27; 6.02) | 1.58 (0.93; 2.70) |
| Use of assist device/ IABP | 4.33 (3.67; 5.09) | 1.08 (0.90; 1.28) | Ref | 2.95 (2.47; 3.52) | 2.63 (2.24; 3.08) |
| CABG | 0.83 (0.75; 0.91) | 0.38 (0.34; 0.41) | Ref | 1.41 (1.30; 1.53) | 1.26 (1.17; 1.35) |
| PCI | 0.34 (0.32; 0.36) | 1.06 (1.02; 1.10) | Ref | 0.53 (0.50; 0.55) | 0.45 (0.43; 0.47) |

All models adjusted for age, sex, race, and latent classes.

* Includes haemorrhagic stroke.

^Class 3 is the largest class and was selected as the reference group.

CABG: coronary artery bypass graft; CHD: coronary heart disease; CKD: chronic kidney disease; COPD: chronic obstructive pulmonary disease; DM: diabetes; HF: heart failure; IABP: intra-aortic balloon pump; PCI: percutaneous coronary intervention; PVD: peripheral vascular disease; VD: valvular disease.

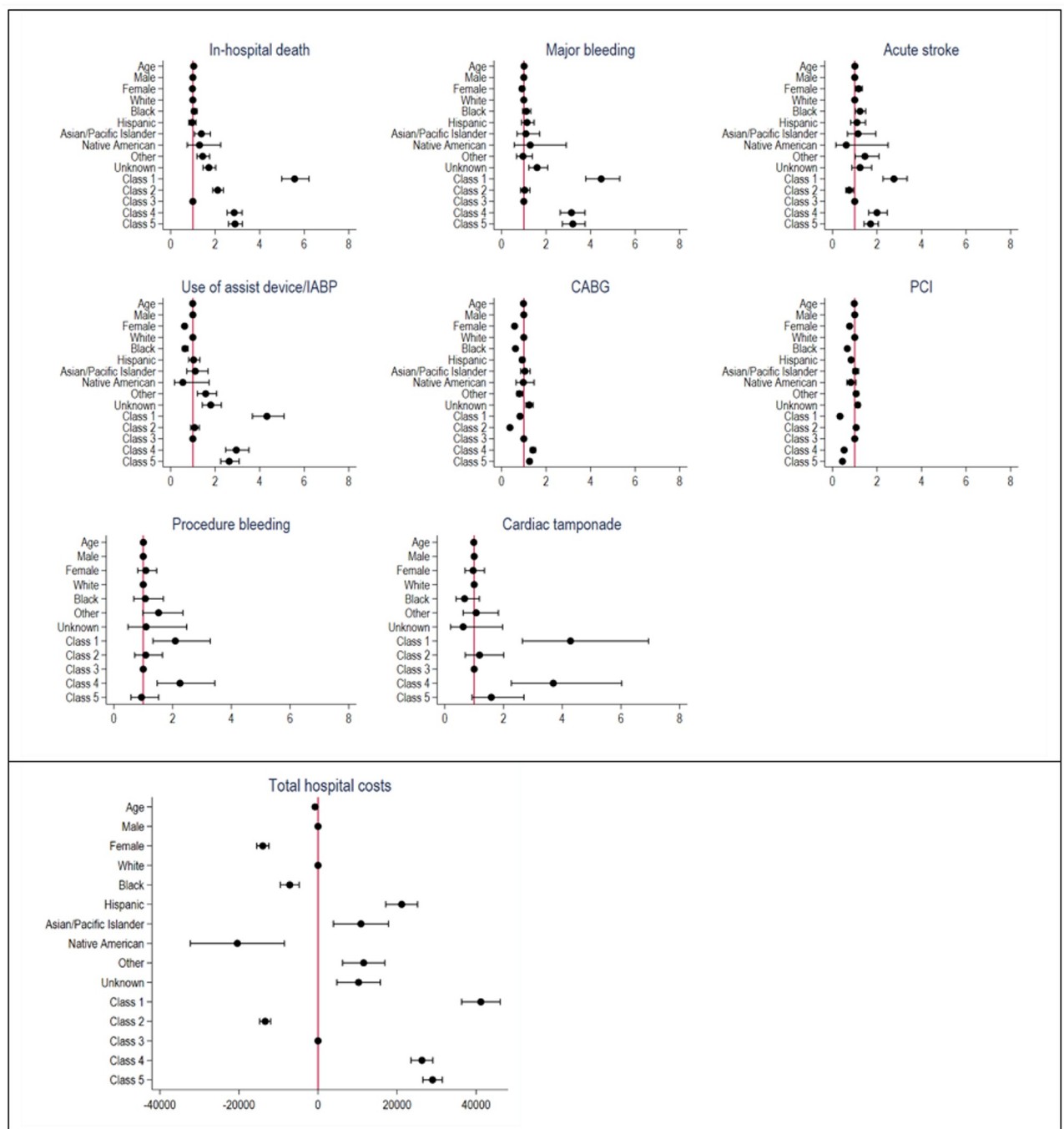

**Fig 3. Forest plot of ORs or regression coefficients (95%CI) for in-hospital outcomes.** CABG: coronary artery bypass graft; IABP: intra-aortic balloon pump; PCI: percutaneous coronary intervention.

found that the most common comorbidities were HTN, CHD, dyslipidaemia, diabetes, HF, chronic pulmonary disease, and CKD. This is in agreement with previous US studies [5, 25–28], Whilst previous work has described the prevalence of comorbidities in patients with CVD, there are only a few studies around how these comorbidities cluster, and whether patient groups with particular clusters of comorbidities have different clinical trajectories/outcomes

[26, 28]. To our knowledge, no prior studies have created latent subgroups in the prediction of adverse outcomes in US people admitted with AMI. Our analysis showed good phenogroup separation of a large cohort of patients hospitalized for AMI into five novel latent classes. We used LCA as it is reportedly superior to other clustering approaches [12], and it has shown to improve the prediction of adverse in outcomes in distinct phenogroups with CVD [29–31]. A large LCA of multimorbidity burden in 693,388 patients admitted with AMI between 2003 and 2013 in England and Wales was based on seven conditions: HTN, diabetes, HF, renal failure, cerebrovascular disease, PVD, and COPD or asthma [32]. Their analysis revealed a high (dominated by CHD and PVD), medium (dominated by PVD and HTN), and low (low prevalence of comorbidities but with PVD) multimorbidity classes. These three classes are not directly comparable to ours given the different included conditions as we examined 21 comorbidities, including mental illness, which further highlights the inclusivity and potential clinical applicability of our findings. However, both studies agree that HTN and diabetes were among the most common comorbidities in AMI admissions. Their study also reports that 59.5% of patients (N = 412,809) had ≥1 comorbidity in comparison to >99% of patients in our study which indicates the different multimorbidity burden between both populations, the more limited number of comorbid conditions that were considered in that analysis, as well as the rapidly increasing prevalence of multimorbidity over time as our later data shows. In China, 75.7% of 49,453 patients with CHD admitted between 2018 and 2020 had ≥1 comorbidity, which is closer to our estimates than UK data [33]. The investigators conducted LCA of 13 comorbidities (CVD, diabetes and Parkinson's disease) which resulted in three comorbidity classes: severe (dominated by HF), moderate (dominated by HTN and diabetes), and mild (fewest comorbidities) which differ from our phenotypes [33].

Examining the clinical and socio-demographic profiles of patients in the emerged classes may inform clinical practice on class membership and why some comorbidities were grouped together. For example, we found that CKD prevalence ranged between 0–35% in Classes 1–4, but was 79% in Class 5, alongside diabetes and HF. One in three adults with diabetes has CKD [34] and type 2 diabetes increases risk of HF development two-fold [35]. Furthermore, approximately 50% of patients with HF have CKD [36], where HF is a risk factor for CKD and vice versa [37–39]. Additionally, other possible determinant of this clustering is that Class 5 have the highest proportion of Black people (17%) who are evidently disproportionally affected by higher burden of diabetes [40, 41], CKD [42], and HF [43] compared to other ethnic groups in the US.

Class 1 includes a higher proportion of females (45%) than Classes 2–5 which is possibly explained by the combination of higher prevalence of comorbidities known to be more common in females, e.g., liver disease [44] (5% in C1 vs. 0.4–2.9% in C2-C5), and hypercoagulation (coagulopathy) [45] (11% vs. 0.7%-8.2% in C2-C5). Class 1 also has the lowest burden of dyslipidaemia (5.7% vs. 33–91% in C2-C5) that is known to be less common in females, possibly due to the reported high prevalence of diagnostic inertia of dyslipidaemia in women [46].

Our findings show that no comorbidity was exclusive per a latent class, especially for the most common comorbidities in AMI population e.g., HF that is most prevalent at 74% in the DM/CKD/HF group (Class 5) while it is also diagnosed in 62% of people in Class 1. Reportedly, HF leads to activation of coagulation (increased levels of thrombin formation and activation of fibrinolysis) [47, 48] and also due to possible cardiohepatic interactions related to the commonly co-existing liver disease and HF [49]. Collectively, these may explain the rates of HF in this Cancer/Coagulopathy/Liver disease dominant group (Class 1).

Patients in Class 4 were the oldest and had the highest prevalence of COPD, VD, and PVD compared to other classes. VD prevalence is significantly higher with increasing age [50], and COPD is prevalent in patients referred to heart valve surgeries [51], whereas PVD is

commonly encountered in patients with COPD due to shared risk factors such as smoking [52]. Our findings are consistent with these epidemiological data, as while COPD, VD, and PVD are also prevalent in classes 1–3, and 5, the clustering analysis determined that patients with AMI and ≥1 of these comorbidities (e.g. COPD) are more similar to patients with the other two comorbidities (VD/PVD) than to patients with COPD, VD, or PVD in classes 1–3, and 5. Hence, Class 4 separated patients with the highest prevalence of these inter-related comorbidities into a distinct phenogroup. In addition, the high prevalence of AF/flutter in Class 4 is likely attributed to the highest prevalence of baseline stroke that often co-exists with AF. AF is an important risk factor for ischaemic stroke [53] as people with AF are up to five times more likely of having stroke [54], which is also shown in S4 Table. Furthermore, ~50% of patients with HF have CKD [36], where HF is a risk factor for CKD and vice versa [37–39]. This may also explain the relatively high prevalence of CKD in Class 4 (35%) that has the second highest proportion of HF. The later observation may also be linked to the fact that CKD is more common in US people aged 65+ years than in younger people [55], where people in Class 4 are the oldest across all classes.

In summary, our findings suggest that the grouping of some comorbidities in specific classes is likely driven by shared pathophysiology or due to health disparities, such as age and race.

**Association between comorbidity classes and in-hospital outcomes.** The presence of ≥1 comorbidity has been associated with adverse outcomes among patients with AMI [25, 26, 56, 57]. However, to the best of our knowledge, no prior US study has assessed whether different clusters of comorbidities impact outcomes differentially. We found that the five comorbidity classes identified had very different outcomes.

*In-hospital mortality*. Patients in Class 1 had the highest odds of in hospital mortality compared to patients in the common Class 3. This observation is likely attributed to that almost a tenth of patients (9.7%) had cancer and that these individuals have poor outcomes [58]. Class 5 included the oldest patients with the highest prevalence of COPD which partially explains their increased odds for in-hospital mortality compared to Class 3. We also observed that the youngest/least burdened Class 2 had higher odds of mortality than Class 3 which likely may be due to that a high proportion of patients in Class 2 present with STEMI 41% or were burdened by other comorbidities (beyond the 21 we studied). A study reported that people in the high (dominated by CHD and PVD) and medium (PVD and HTN) comorbidity classes in UK AMI admissions between 2003 and 2013 were at 2.4-fold (95%CI: 2.3–2.5) and 1.5-fold (95% CI 1.4±1.5) higher risk of all-cause death compared to the low class, respectively [32]. Similarly, people with CHD in the severe (dominated by HF) and moderate (HTN and DM) comorbidity classes in China were associated with greater risks of mortality and rehospitalization than those in the mild class [33]. In comparison, we found higher mortality risk in the least burdened class as explained above, but we included far more comorbidities and investigated more outcomes besides mortality.

*Major bleeding and stroke*. In comparison to patients in the 'common' group (Class 3), all classes were at higher risk for bleeding and acute stroke outcomes except the youngest/least burdened patients (Class 2). Patients in the cancer/coagulopathy/liver disease group had the worse outcomes and had 4.5-fold higher odds of having a major bleeding episode (which includes haemorrhagic stroke events) and 2.8-fold higher odds for ischemic stroke. Reportedly, coagulopathies, anemia, and thrombocytopenia associated with cancers increases the risk of major bleeding complications in AMI [58]. Our finding is also in line with several studies reporting that liver disease increases the risk for overall, ischemic/haemorrhagic stroke and that most people with liver cirrhosis have coagulopathy, dyslipidaemia, and heart disease, which are associated with developing stroke [59, 60].

*PCI*. Members of the highest comorbidity burden group (diabetes/CKD/HF, Class 5) were less likely to be treated with PCI than members of the 'common' group (Class 3). Reportedly, patients admitted with AMI with higher comorbidity burden are less likely to have cardiac catheterisation [25, 27].

## Potential clinical and research implications

In current clinical practice, patients with CVD without comorbidities are increasingly an exception. Based on contemporary US data, our detailed mapping of comorbidity profile by patient characteristics and association with outcomes improves the understanding of AMI multimorbidity burden which has possible implications in supporting clinicians, wider health-care professionals (HCPs), and policy makers towards developing timely and tailored screening, prevention, and care programmes of these comorbidities in people with AMI. These implications include helping clinicians to identify higher-risk patients with AMI with specific multimorbidity phenotypes as a starting point to assess personalized interventions to reduce premature mortality and possibly prioritising interventions for those at greatest risk of poor outcomes, and defining multidisciplinary therapeutic pathways and novel technologies tailored to the multimorbidity burden in these specific groups. In addition, our findings on the associations of some comorbidity classes with worse outcomes, and our hypotheses as to why certain comorbidities are grouped together (e.g. the clustering of diabetes, CKD, and HF in the Black-dominant Class 5), highlight that prevention and timely screening programmes of some conditions may help in early detection and better management of these comorbidities, in people with AMI. Our identification of higher risk phenotypes may be useful in highlighting patients that may have greatest benefit from early outpatient follow up, and involvement of multi-speciality teams for inpatient or follow-up care. Furthermore, our defined phenotypes can also inform developers of clinical guidelines to consider multimorbidity in their decision-making algorithms [5], primarily in relation to the use of invasive treatments (e.g. NSTEMI), where there may be equipoise, particularly in elderly comorbid patients, thus moving away from current classic individual-condition guidelines.

Areas for future work include the external validation of our findings in a distinct and independent cohort, developing more inclusive research study designs to accommodate people with multimorbidity, examining longitudinal multimorbidity patterns, investigating why comorbidities cluster and how non-cardiovascular comorbidities possibly interact with outcomes in AMI.

## Strengths and limitations

Our analysis is based on mapping the comorbidity burden of a cohort of nearly half-million AMI US admissions using 21 physical and mental health conditions from a nationally representative sample of the US population. To our best knowledge, this is the first study to examine the comorbidity burden of US AMI hospitalizations by patient factors using LCA and assess its association with outcomes. LCA is reportedly an optimum multimorbidity clustering algorithm and superior when compared to other existing clustering methods with many advantages e.g., accommodating different types of data, having a high within-method repeatability, and uniquely identifying small comorbidity clusters possibly not identified by most other clustering methods [12]. In addition, applying sampling weights produced estimates that are generalizable to a larger population than the studied sample size, but not necessarily to all patients with AMI. Therefore, our findings are likely broadly generalizable given the size and the ethnic diversity of the patient cohort and may have possible implications for care for wider population of multimorbid patients with AMI who are often excluded from clinical trials. Our study

has the following limitations. The diagnoses are based on recorded codes, where there is a possibility of inaccurate coding. The dataset lack information on timing of events or in-hospital outcomes and on the underlying cause of death. The analysed data may include recurrent admissions, which cannot be identified, driven by the NIS design based on admission record as the unit of analysis. We have not externally validated the results, but this has been highlighted as a priority for future work. Finally, we may have missed other comorbidities, but we aimed to include AMI-related and AHRQ/ELIXHAUSER-verified comorbidities [20] to minimise misclassification.

## Conclusions

Patients admitted with AMI have a high comorbidity burden, including non-cardiometabolic comorbidities, which we defined into five distinct phenogroups that are highly predictive of specific adverse outcomes. The grouping of some comorbidities in specific classes is likely driven by shared pathophysiology or due to health disparities, at least partially driven by age and race. The phenogroups we identified have different mortality, bleeding, and stroke outcomes, and we also observed heterogeneity in revascularisation. Our findings agree with the current trend towards prioritising and assessing multimorbidity in patients admitted with AMI, and CVD overall, to better understand the care needs of patients beyond the recommendations from the conventional single-disease oriented guidelines.

## Supporting information

**S1 Fig. Latent class membership (percentage) by age, sex, age/sex, and race.** Radar charts present the percentage proportion of patients per latent class starting from 0% (chart centre). (PDF)

**S2 Fig. Individual radar charts for percentage prevalence of comorbidities per latent class.** (PDF)

**S3 Fig. Radar charts for percentage prevalence of in-hospital outcomes per latent class.** (PDF)

**S4 Fig. Predictive margins (95% CI) of probabilities of outcomes per latent class.** (PDF)

**S1 Table. ICD-10 and procedure codes for outcomes.** (PDF)

**S2 Table. Comparison of AIC/BIC values between models.** (PDF)

**S3 Table. Significance testing for baseline characteristics and comorbidity burden across latent classes.** (PDF)

**S4 Table. Crude rates of in-hospital outcomes in patients admitted with AMI in 2018.** (PDF)

**S5 Table. Odds ratios (95% CI) for risks of in-hospital outcomes in patients admitted with AMI in 2018.** * Includes haemorrhagic stroke. ^ Class 3 is the largest class and was selected as the reference group. $ The 'Other' race group for these outcomes includes Hispanic, Asian/ Pacific Islander, Native American and 'Other' categories. CABG: coronary artery bypass graft; CHD: coronary heart disease; DM: diabetes; CKD: chronic kidney disease; COPD: chronic

obstructive pulmonary disease; HF: heart failure; IABP: intra-aortic balloon pump; PCI: percutaneous coronary intervention; PVD: peripheral vascular disease; VD: valvular disease.
(PDF)

**S6 Table. Regression coefficients (95% CI) of predictors of hospital costs in patients admitted with AMI in 2018.** ^ Class 3 is the largest class and was selected as the reference group. CHD: coronary heart disease; DM: diabetes; CKD: chronic kidney disease; COPD: chronic obstructive pulmonary disease; HF: heart failure; PVD: peripheral vascular disease; VD: valvular disease.
(PDF)

**S7 Table. Incidence rate ratios (IRRs) (95% CI) of predictors of length of hospital stay in patients admitted with AMI in 2018.** ^ Class 3 is the largest class and was selected as the reference group. CHD: coronary heart disease; DM: diabetes; CKD: chronic kidney disease; COPD: chronic obstructive pulmonary disease; HF: heart failure; PVD: peripheral vascular disease; VD: valvular disease.
(PDF)

## Author Contributions

**Conceptualization:** Salwa S. Zghebi.

**Data curation:** Salwa S. Zghebi, Muhammad Rashid.

**Formal analysis:** Salwa S. Zghebi.

**Funding acquisition:** Salwa S. Zghebi.

**Project administration:** Salwa S. Zghebi.

**Supervision:** Evangelos Kontopantelis, Mamas A. Mamas.

**Writing – original draft:** Salwa S. Zghebi.

**Writing – review & editing:** Salwa S. Zghebi, Martin K. Rutter, Louise Y. Sun, Waqas Ullah, Muhammad Rashid, Darren M. Ashcroft, Douglas T. Steinke, Stephen Weng, Evangelos Kontopantelis, Mamas A. Mamas.

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
