## [Decision Letter · Decision Letter 0]

18 Sep 2023

PONE-D-23-21645Comorbidity clusters and in-hospital outcomes in patients admitted with acute myocardial infarction in the USA: a national population-based studyPLOS ONE

Dear Dr. Zghebi,

Thank you for submitting your manuscript to PLOS ONE. After careful consideration, we feel that it has merit but does not fully meet PLOS ONE’s publication criteria as it currently stands. Therefore, we invite you to submit a revised version of the manuscript that addresses the points raised during the review process.

We look forward to receiving your revised manuscript.

Kind regards,

Amirmohammad Khalaji

Academic Editor

PLOS ONE

“SSZ, LYS, EK, MKR, DS, DMA MAM, MR declare no competing interests. SW is a currently an employee of GSK.”

Reviewers' comments:

Reviewer's Responses to Questions

**Comments to the Author**

1. Is the manuscript technically sound, and do the data support the conclusions?

Reviewer #1: Yes

Reviewer #2: Yes

2. Has the statistical analysis been performed appropriately and rigorously? 

Reviewer #1: I Don't Know

Reviewer #2: Yes

3. Have the authors made all data underlying the findings in their manuscript fully available?

Reviewer #1: Yes

Reviewer #2: Yes

4. Is the manuscript presented in an intelligible fashion and written in standard English?

Reviewer #1: Yes

Reviewer #2: Yes

5. Review Comments to the Author

Reviewer #1: Zghebi et al. have performed a population-based study in order to identify comorbidity phenotypes of worse clinical outcomes following acute MI. The study is well performed and the findings are interesting. There are some minor points to add.

- The authors are encouraged to explain the clinical implications of their findings. In other words, what are the main lessons a clinician can learn from these?

- The advantages of LCA over other clustering methods could be added to the discussion section.

Reviewer #2: Few Questions:

1. How do authors explain high morbidity in Class 2 least burdened group as compared to Class 3 (CHD/dyslipidemia) group.

2. What explains the higher proportion of female in the Class 1 cluster?

3. What is the explanation of higher preponderance of afib/flutter in Class 4 patients.

4. What explains higher CKD rates in Class 4 patients?

5. One would imagine higher HF rates in CHD class, and yet HF rates are towards the lower side in Class 3 compared to Class 1, 4 and 5

6. What explains higher HF rates in Class 1

6. PLOS authors have the option to publish the peer review history of their article (what does this mean?). If published, this will include your full peer review and any attached files.

Reviewer #1: No

Reviewer #2: **Yes: **Nadeem Afridi

---

## [Author Response · Author response to Decision Letter 0]

3 Oct 2023

Dear Editor,

Re. PONE-D-23-21645

Thank you for the opportunity to submit a revision of our manuscript. 

Below, we provide a point-by-point response to the received comments. The changes are highlighted on the revised manuscript and numbered by the Reviewer and comment number to facilitate the review process. 

We hope to have satisfactorily responded to the comments, which have strengthened our manuscript.

Looking forward to receiving your decision.

Yours sincerely,

Dr Zghebi, on behalf of the authors.

Journal requirements

https://journals.plos.org/plosone/s/file?id=wjVg/PLOSOne_formatting_sample_main_body.pdf [journals.plos.org] and

https://journals.plos.org/plosone/s/file?id=ba62/PLOSOne_formatting_sample_title_authors_affiliations.pdf [journals.plos.org]

Authors response

We thank the Editor for this comment. The manuscript has been reviewed and the formatting now adheres to the Journal's style as advised. 

“SSZ, LYS, EK, MKR, DS, DMA MAM, MR declare no competing interests. SW is a currently an employee of GSK.” Please confirm that this does not alter your adherence to all PLOS ONE policies on sharing data and materials, by including the following statement: "This does not alter our adherence to PLOS ONE policies on sharing data and materials.” (as detailed online in our guide for authors http://journals.plos.org/plosone/s/competing-interests). [journals.plos.org] If there are restrictions on sharing of data and/or materials, please state these. Please note that we cannot proceed with consideration of your article until this information has been declared. Please include your updated Competing Interests statement in your cover letter; we will change the online submission form on your behalf.

Authors response

We confirm that the declared competing Interests statement does not alter our adherence to PLOS ONE policies on sharing data and materials. The updated Competing Interests statement has been added to the cover letter, as requested. 

3. In your Data Availability statement, you have not specified where the minimal data set underlying the results described in your manuscript can be found. PLOS defines a study's minimal data set as the underlying data used to reach the conclusions drawn in the manuscript and any additional data required to replicate the reported study findings in their entirety. All PLOS journals require that the minimal data set be made fully available. For more information about our data policy, please see http://journals.plos.org/plosone/s/data-availability [journals.plos.org].

Upon re-submitting your revised manuscript, please upload your study’s minimal underlying data set as either Supporting Information files or to a stable, public repository and include the relevant URLs, DOIs, or accession numbers within your revised cover letter. For a list of acceptable repositories, please see http://journals.plos.org/plosone/s/data-availability#loc-recommended-repositories [journals.plos.org]. Any potentially identifying patient information must be fully anonymized.

Important: If there are ethical or legal restrictions to sharing your data publicly, please explain these restrictions in detail. Please see our guidelines for more information on what we consider unacceptable restrictions to publicly sharing data: http://journals.plos.org/plosone/s/data-availability#loc-unacceptable-data-access-restrictions [journals.plos.org]. Note that it is not acceptable for the authors to be the sole named individuals responsible for ensuring data access. We will update your Data Availability statement to reflect the information you provide in your cover letter.

Authors response

An updated Data Availability statement has been added to the cover letter as requested. Also copied below:

The data underlying the results presented in the study are available for purchase from the Healthcare Cost and Utilization Project (HCUP) - National (Nationwide) Inpatient Sample (NIS) via their website https://www.hcup-us.ahrq.gov/nisoverview.jsp. All interested researchers can access the data through HCUP directly and authors are not permitted to share the data (even as minimal underlying dataset) or make it available as per the data use agreement with HCUP. The authors did not have any special access privileges to this data. 

Authors response

We confirm that the reference list is complete and correct, and we have not cited any retracted references. New references numbered 44 to 49 and 53 to 55 have been added to the revised manuscript as part of the response to the reviewers' comments.

Review Comments to the Author

Reviewer #1

Zghebi et al. have performed a population-based study in order to identify comorbidity phenotypes of worse clinical outcomes following acute MI. The study is well performed and the findings are interesting. There are some minor points to add.

1. The authors are encouraged to explain the clinical implications of their findings. In other words, what are the main lessons a clinician can learn from these?

Authors response

We thank the reviewer for the positive remarks on our study and findings. 

The findings provide a new insight on the clustering of comorbidity burden in patients admitted with AMI which can help clinicians to identify higher-risk patients with AMI with specific multimorbidity phenotypes as a starting point to assess personalized interventions to minimise adverse outcomes and reduce premature mortality. 

Alongside our mapping of comorbidity profile, we provide an interpretation of possibly why certain comorbidities grouped together in people with AMI which may support healthcare professionals (HCPs) towards developing tailored prevention and timely screening programmes of some conditions may help early detection and better management of these comorbidities in people with AMI.

As outlined in the revised manuscript, we observe the following main lessons from our findings to clinicians, wider HCPs and policy makers:

- Understanding of the detailed mapping of the comorbidity profile by patient factors in adults admitted with AMI derived from modern clinical data. 

- Understanding of possibly why certain comorbidities group together in people with AMI which can inform screening schemes and the adoption of alternative multidisciplinary therapeutic pathways tailored for high-risk groups with a specific multimorbidity phenotype e.g. our interpretation of the drivers of COPD, VD, and PVD clustering in Class 4, or the clustering of DM, CKD, and HF in the Black-dominant Class 5.

- Inform clinical guidelines and healthcare policy developers on class membership to move away from current individual-condition guideline.

Changes to the paper

In response to this comment, we have revised the 'Potential clinical and research implications' section of the Discussion and it now reads as follows:

'In current clinical practice, patients with CVD without comorbidities are increasingly an exception. Based on contemporary US data, our detailed mapping of comorbidity profile by patient characteristics and association with outcomes improves the understanding of AMI multimorbidity burden which has possible implications in supporting clinicians, wider healthcare professionals (HCPs), and policy makers towards developing timely and tailored screening, prevention, and care programmes of these comorbidities in people with AMI. These implications include helping clinicians to identify higher-risk patients with AMI with specific multimorbidity phenotypes as a starting point to assess personalized interventions to reduce premature mortality and possibly prioritising interventions for those at greatest risk of poor outcomes, and defining multidisciplinary therapeutic pathways and novel technologies tailored to the multimorbidity burden in these specific groups. In addition, our findings on the associations of some comorbidity classes with worse outcomes, and our hypotheses as to why certain comorbidities are grouped together (e.g. the clustering of diabetes, CKD, and HF in the Black-dominant Class 5), highlight that prevention and timely screening programmes of some conditions may help in early detection and better management of these comorbidities, in people with AMI. Our identification of higher risk phenotypes may be useful in highlighting patients that may have greatest benefit from early outpatient follow up, and involvement of multi-speciality teams for inpatient or follow-up care. Furthermore, our defined phenotypes can also inform developers of clinical guidelines to consider multimorbidity in their decision-making algorithms,[5] primarily in relation to the use of invasive treatments (e.g. NSTEMI), where there may be equipoise, particularly in elderly comorbid patients, thus moving away from current classic individual-condition guidelines. 

Areas for future work include the external validation of our findings in a distinct and independent cohort, developing more inclusive research study designs to accommodate people with multimorbidity, examining longitudinal multimorbidity patterns, understanding why comorbidities cluster and how non-cardiovascular comorbidities possibly interact with outcomes in AMI.'

2. The advantages of LCA over other clustering methods could be added to the discussion section.

Authors response and Changes to the paper

In response to this comment, we now added the following section to the Discussion section:

'LCA is reportedly an optimum multimorbidity clustering algorithm and superior when compared to other existing clustering methods with many advantages e.g., accommodating different types of data, having a high within-method repeatability, and uniquely identifying small comorbidity clusters possibly not identified by most other clustering methods.[12]'

Reviewer #2

Few Questions:

1. How do authors explain high morbidity in Class 2 least burdened group as compared to Class 3 (CHD/dyslipidemia) group.

Authors response

As shown in Table 2 and S2 Figure (both copied in below), we observe that Class 2 has a consistently lower morbidity burden compared to people in Class 3, apart from 'rheumatoid arthritis and collagen vascular diseases' where the proportion is at 1.9% in both classes and weight loss that is more common in C2. Notably, in some conditions the burden is much lower in Class 2 e.g., for HTN (40% vs. 95%), dyslipidaemia (35% vs. 93%), diabetes (6% vs. 51%), and CKD (0% vs. 6%).

Comorbidities, % proportion (95% CI) Class 2 (Least burdened) Class 3 (CHD/ dyslipidaemia)

Hypertension 40.4 (39.7; 41.2) 95.0 (94.8; 95.2)

CHD 67.2 (66.5; 67.9) 90.6 (90.3; 90.9)

Dyslipidaemia 35.2 (34.5; 35.9) 92.6 (92.3; 92.9)

Diabetes (DM) 5.8 (5.5; 6.2) 51.2 (50.6; 51.7)

Heart failure (HF) 5.5 (5.1; 5.8) 18.1 (17.7; 18.5)

COPD 11.3 (10.8; 11.8) 16.9 (16.5; 17.3)

CKD 0 5.9 (5.7; 6.2)

Obesity 10.2 (9.7; 10.7) 29.2 (28.7; 29.6)

Anaemias 2.7 (2.5; 3.0) 3.6 (3.4; 3.8)

Valvular disease (VD) 5.5 (5.2; 5.8) 7.0 (6.7; 7.2)

Hypothyroidism 7.2 (6.8; 7.6) 11.0 (10.6; 11.3)

Atrial fibrillation (AF)/flutter 3.3 (3.0; 3.5) 5.1 (4.9; 5.3)

PVD 3.2 (2.9; 3.4) 4.5 (4.3; 4.7)

Depression 5.0 (4.7; 5.4) 9.9 (9.6; 10.2)

TIA/Stroke 1.7 (1.5; 1.9) 6.6 (6.31; 6.8)

Coagulopathy 1.1 (1.0; 1.3) 3.1 (2.90; 3.3)

Weight loss 0.4 (0.4; 0.6) 0.2 (0.2; 0.23)

Cancer 1.0 (0.8; 1.1) 1.2 (1.1; 1.3)

RA & collagen vascular diseases 1.9 (1.7; 2.1) 1.9 (1.7; 2.0)

Liver disease 0.5 (0.4; 0.6) 1.8 (1.7; 2.0)

Psychoses 1.3 (1.2; 1.5) 2.2 (2.0; 2.3)

2. What explains the higher proportion of female in the Class 1 cluster?

Authors response

Overall, Class 1 is dominated by a higher prevalence of liver disease, coagulopathy, and cancer, compared to other classes. Previous studies show that female sex is associated with higher risk of acute on chronic liver failure (ref 44), and with hypercoagulation (ref 45), which possibly explains why Class 1 had more females (45%) than other classes (proportion ranging from 34% in Class 2 to 43% in Class 4). 

As shown in the female-specific heat map (Figure 2), Class 1 combines higher prevalence of comorbidities known to be more common in females: liver disease (5% in C1 vs. 0.4-2.9% in C2-C5), coagulopathy (11% vs. 0.7%-8.2% in C2-C5), and is the class with the lowest burden of dyslipidaemia (5.7% vs. 33-91%) that is known to be less common in females, possibly due to the reported high prevalence of diagnostic inertia of dyslipidaemia in women (ref 46).

Changes to the paper

In response to this comment, the following text is now added to the Discussion section: 'Class 1 includes a higher proportion of females (45%) than Classes 2-5 which is possibly explained by the combination of higher prevalence of comorbidities known to be more common in females, e.g., liver disease[44] (5% in C1 vs. 0.4-2.9% in C2-C5), and hypercoagulation (coagulopathy)[45] (11% vs. 0.7%-8.2% in C2-C5). Class 1 also has the lowest burden of dyslipidaemia (5.7% vs. 33-91% in C2-C5) that is known to be less common in females, possibly due to the reported high prevalence of diagnostic inertia of dyslipidaemia in women.[46]'

3. What is the explanation of higher preponderance of afib/flutter in Class 4 patients.

Authors response

The high prevalence of AF/flutter in Class 4 (28%) is likely attributed to the highest prevalence of baseline stroke (18%) that often co-exists with AF. It is widely known that AF is an important risk factor for ischaemic stroke (ref 53) as people with AF are up to five times more likely of having stroke (ref 54). This is also evident by the results in S4 Table showing that Class 4 members are the top 2 among all classes in developing acute ischemic stroke outcome (1.7%). 

Changes to the paper

In response to this comment, this text has been added to the Discussion section of the paper: 'In addition, the high prevalence of AF/flutter in Class 4 is likely attributed to the highest prevalence of baseline stroke that often co-exists with AF. AF is an important risk factor for ischaemic stroke[53] as people with AF are up to five times more likely of having stroke,[54] which is also shown in S4 Table.'

4. What explains higher CKD rates in Class 4 patients?

Authors response

The rates of CKD in Class 4 (35%) are probably linked to the fact that CKD is generally prevalent in adults with CVD, hence its prevalence ranges between 0-25% in Classes 1-3 and up to 79% in Class 5. Previous literature show that ~50% of patients with HF have CKD, [36] where HF is a risk factor for CKD and vice versa, [37-39] where Class 4 has the second highest proportion of HF.

Age may also explain the observed higher CKD rates in Class 4. As highlighted in the paper, people in Class 4 are the oldest group compared to the other classes and The Centers for Disease Control and Prevention (CDC) report that CKD is more common in US people aged 65+ years (34%) than in younger people (ref 55) which is in agreement with our estimates. 

Changes to the paper

In response to this comment, the following text in the Discussion section now reads as: 'Furthermore, ~50% of patients with HF have CKD,[36] where HF is a risk factor for CKD and vice versa.[37-39] This may also explain the relatively high prevalence of CKD in Class 4 (35%) that has the second highest proportion of HF. The later observation may also be linked to the fact that CKD is more common in US people aged 65+ years than in younger people,[55] where people in Class 4 are the oldest across all classes.'

5. One would imagine higher HF rates in CHD class, and yet HF rates are towards the lower side in Class 3 compared to Class 1, 4 and 5

Authors response

The chosen clustering approach expectedly classifies the patient population into distinct phenogroups based on the more relevant or inter-related comorbidities (e.g. conditions with shared pathophysiology) and socio-demographic factors (and possibly other unrecorded characteristics). In other words, LCA determined that patients with AMI and with high rates of HF are likely 'more similar' to AMI patients with highly prevalent diabetes and CKD (and so were grouped in Class 5) than to AMI patients with prevalent CHD and dyslipidaemia in Class 3. However, as the reviewer noted, HF is prevalent in Class 1 and Classes 3-5, but the resulted LCA phenogroups are driven by several factors as we explain above. 

6. What explains higher HF rates in Class 1

Authors response

Our findings show that no comorbidity was exclusive per a latent class, especially for the most common comorbidities in AMI population e.g. HTN, HF and CHD. While we acknowledge that LCA grouped people with AMI into the most distinct clusters as possible leading to some clusters with a few dominant comorbidities, the observation of high HF rates in Class 1 may be basically driven by HF being a common comorbidity in people with AMI and so it is most prevalent at 74% in the DM/CKD/HF group (Class 5), while is also diagnosed in 62% of people in Class 1. Reportedly, HF leads to activation of coagulation (increased levels of thrombin formation and activation of fibrinolysis) (ref 47 & 48) and also due to possible cardiohepatic interactions related to the commonly co-existing liver disease and heart failure. (ref 49) Collectively, these may explain the observed rates of HF in this Cancer/Coagulopathy/Liver disease dominant group (Class 1).

Changes to the paper

This text has been added to the Discussion section to explain this observation: 'Our findings show that no comorbidity was exclusive per a latent class, especially for the most common comorbidities in AMI population e.g., HF that is most prevalent at 74% in the DM/CKD/HF group (Class 5) while it is also diagnosed in 62% of people in Class 1. Reportedly, HF leads to activation of coagulation (increased levels of thrombin formation and activation of fibrinolysis)[47, 48] and also due to possible cardiohepatic interactions related to the commonly co-existing liver disease and HF.[49] Collectively, these may explain the rates of HF in this Cancer/Coagulopathy/Liver disease dominant group (Class 1).'

---

## [Editor Report · Decision Letter 1]

10 Oct 2023

Comorbidity clusters and in-hospital outcomes in patients admitted with acute myocardial infarction in the USA: a national population-based study

PONE-D-23-21645R1

Dear Dr. Zghebi,

We’re pleased to inform you that your manuscript has been judged scientifically suitable for publication and will be formally accepted for publication once it meets all outstanding technical requirements.

Kind regards,

Amirmohammad Khalaji

Academic Editor

PLOS ONE
---

## [Editor Report · Acceptance letter]

16 Oct 2023

PONE-D-23-21645R1 

Comorbidity clusters and in-hospital outcomes in patients admitted with acute myocardial infarction in the USA: a national population-based study 

Dear Dr. Zghebi:

I'm pleased to inform you that your manuscript has been deemed suitable for publication in PLOS ONE. Congratulations! Your manuscript is now with our production department. 

Kind regards, 

on behalf of

Dr. Amirmohammad Khalaji 

Academic Editor

PLOS ONE